# A Path Tracking Method of a Wall-Climbing Robot towards Autonomous Inspection of Steel Box Girder

**Wei Song** [1,2,3] **, Zhijian Wang** [1] **, Tong Wang** [1] **, Daxiong Ji** [1,*] **and Shiqiang Zhu** [1,2]

1   Ocean College, Zhejiang University, Zhoushan 316000, China; weisong@zju.edu.cn (W.S.);
    21834119@zju.edu.cn (Z.W.); 3160100561@zju.edu.cn (T.W.); sqzhu@sfp.zju.edu.cn (S.Z.)
2   Zhejiang Lab, Hangzhou 311100, China
3   Technology and Equipment of Rail Transit Operation and Maintenance Key Laboratory of Sichuan Province,
    Southwest Jiaotong University, Chengdu 610031, China
*   Correspondence: jidaxiong@zju.edu.cn; Tel.: +86-18334324673

**Abstract:** This paper proposes an autonomous inspection method for steel box girders that uses a wall-climbing robot instead of human workers. According to the 3D operating environment inside the steel box girder, the method proposes a 3D path for the robot to traverse positions to be inspected. The path comprises two alternate sections of the lane's centerline and U-shaped steering. To realize the robot's tracking of the desired path, kinematics analysis based on different poses during the inspection was carried out. Corresponding path tracking algorithms were adopted to ensure that the robot moves accurately and efficiently. In addition, for the smooth transition of the two path sections, this method adopts an algorithm of cooperatively controlling the lifting mechanism and the wheel speeds to achieve stable crossing of a 90° concave corner. Finally, experiment results show that the robot can steadily cross 90° concave corner and can steer to the adjacent lane and complete lane inspection along the desired path. The method can realize autonomous inspection for steel box girders using the wall-climbing robot.

**Keywords:** steel box girder; wall-climbing robot; inspection path; path tracking

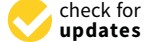


## 1. Introduction

Steel box girder (SBG) is an important structural unit of long-span bridges, and its condition inspection is an important means to ensure safe use. SBG is an approximately closed structure, and its inspection is usually aimed at the internal structure. As shown in Figure 1, the current detection methods are mainly manual [1,2]. Workers use ladders to reach the target wall and then use naked eyes or flaw detection equipment to perform an inspection. This method has limitations such as low detection efficiency, high labor intensity, and a certain degree of danger. However, the wall-climbing robot, as a mobile platform that can be attached to the wall, can replace workers to complete labor-intensive and high-risk tasks.

To overcome the shortcomings of manual inspection, researchers have developed many wall-climbing robots that can be used in bridge maintenance and inspection [3–11]. For example, the biomimetic inchworm robot in reference [3] can transition between two walls with an angle of 0–360°, and can pass through the manhole of the bridge partition; the tracked wall-climbing robot in reference [10] adapts to the cylindrical wall surface of the bridge by using a reciprocating mechanism. These robots operate on multiple planes or curved surfaces, which means relatively complex work paths and difficult path tracking. To conduct a comprehensive and reliable inspection, the primary goal of this task is to ensure that the wall-climbing robot accurately tracks the inspection path. Therefore, it is essential to perform effective path tracking for the robot.

Nowadays, path tracking algorithms are mainly based on geometric models [12–14], kinematic models [15,16], and dynamic models [17–20]. In our study, the robot moves

with low acceleration, low speed and light weight, so the path tracking control based on the dynamic model was not exploited. Among tracking algorithms based on a geometric model, Wallace et al. [12] used image processing technology to extract the centerline of the road boundary. They kept the vehicle "watching" a position on the centerline at a certain distance, then calculated the distance deviation between the vehicle and the centerline, thereby controlling the vehicle to travel along the road's centerline. Ma et al. [13] combined a fuzzy controller with a pure tracking algorithm and used driving experience to dynamically formulate fuzzy control rules to adjust the forward-looking distance. Their simulation results show that the algorithm improves the performance of path tracking. Among tracking algorithms based on a kinematics model, the authors in [15] established a kinematic model for the path tracking of a robot. It introduces "virtual control variables" to decompose complex nonlinear systems into low-order systems, then constructs a Lyapunov function to analyze the system's stability, and finally derives the control law to track the desired path. O'Connor at Stanford University [16] established a kinematic model that uses the rate of change of heading deviation, lateral deviation, and front-wheel angle as state variables. Then, they linearized the model to design a path tracking controller. These path tracking algorithms are characterized by a simple implementation process, easy adjustment of parameters, and low amount of calculations. They are widely used because they are especially suitable for low-speed working conditions [14]. However, due to the strong magnetic field, weak GPS signal, and special three-dimensional structure of the working conditions, the above-mentioned path tracking algorithm is not suitable for SBG inspection. In addition, in this scenario, the wall-climbing robot must perform corresponding tracking algorithms on different walls and ensure that it transitions smoothly and steadily between different walls.

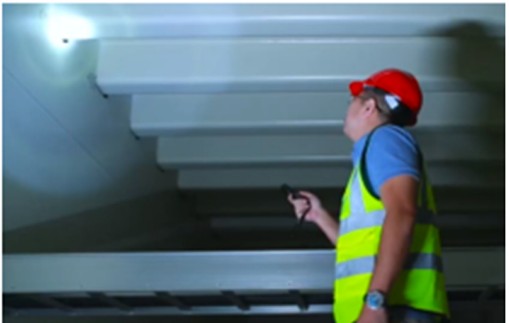 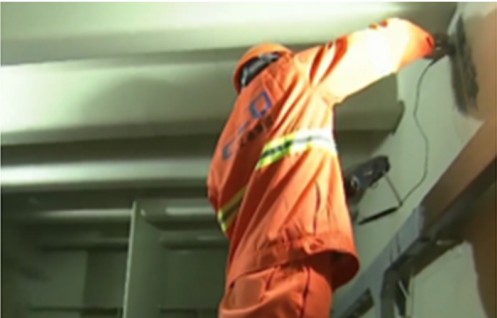

**Figure 1.** Manual inspection inside the SBG of a bridge.

This paper proposes a method suitable for autonomous inspection in SBG to address the above problems. First, the prototype of the wall-climbing robot is introduced. The inspection path is divided into two alternate sections of the lane's centerline and U-shaped steering. The coordinated control algorithm of the lifting mechanism and the wheel speeds is proposed to perform an autonomous transition between the two sections. A "Preview-follow" algorithm is adopted to track the lane's centerline accurately. A dead reckoning algorithm is used to track the U-shaped path with slippage restrained. Finally, the experiment of autonomous inspection in the SBG scene was carried out to verify the effectiveness of these algorithms adopted by our method.

## 2. Prototype

The prototype of the wall-climbing robot is shown in Figure 2, and its internal structure is shown in Figure 3. The actual weight of the prototype is 4.08 kg, and the size is 230 mm × 210 mm × 256 mm. The robot uses four permanent magnet wheels to achieve stable adsorption on the steel wall. The advantage of the magnetic wheel is that it has strong adaptability to the wall surface and is easy to control. However, the wheeled structure's ability to overcome obstacles is relatively limited. For example, this robot cannot cross convex corners.

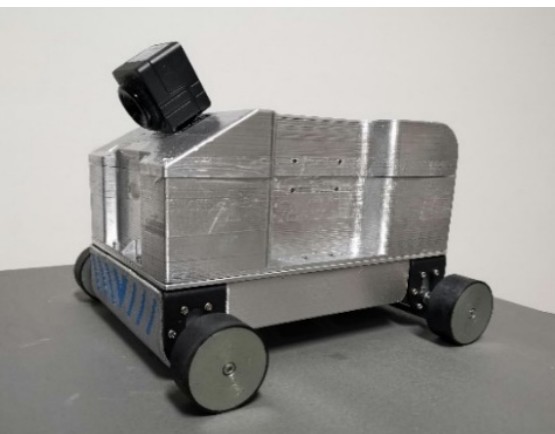

**Figure 2.** Prototype of the wall-climbing robot.

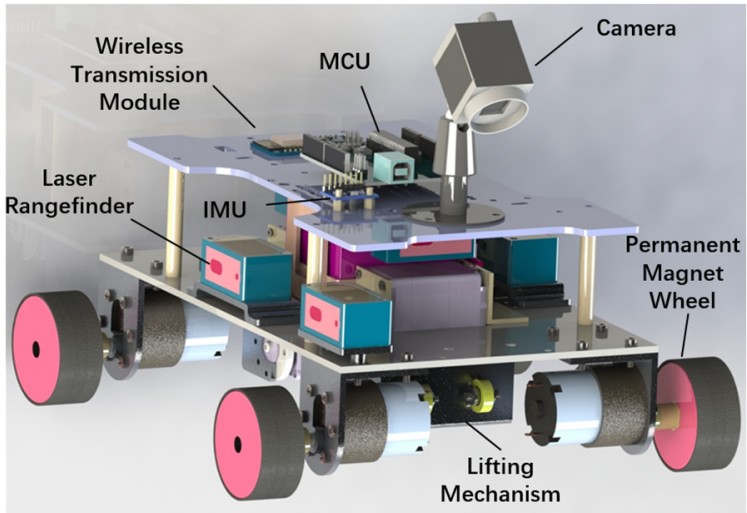

**Figure 3.** The internal structure of the wall-climbing robot.

The robot uses a lifting mechanism driven by the steering gear to help the magnetic wheel climb up the vertical wall. Furthermore, to simplify the structure and reduce the self-weight, the robot has no Ackerman steering mechanism; it uses the differential drive method to steer. The robot also carries a camera to take photos of the lane, which are transmitted to a computer for real-time display. When working in the SBG, it will traverse every lane for the staff to inspect.

## 3. Modeling and Solution

The focus of the SBG inspection is the weld between the roof deck and the U-rib. The roof deck and two adjacent U-ribs form a patrol lane. The desired tracking path comprises paralleled centerlines of the lane on the roof and U-shaped paths on the crossbeam (as shown in Figure 4a). The robot completed three actions along the path (as shown in Figure 4b): crossing a 90° concave corner, tracking the lane's centerline, and steering to the adjacent lane. To ensure accurate and reliable path tracking, the tracking algorithms should make the response fast and the error small, and crossing concave corners should be sufficiently smooth and stable. Accordingly, our method adopts corresponding motion control algorithms for the three different actions.

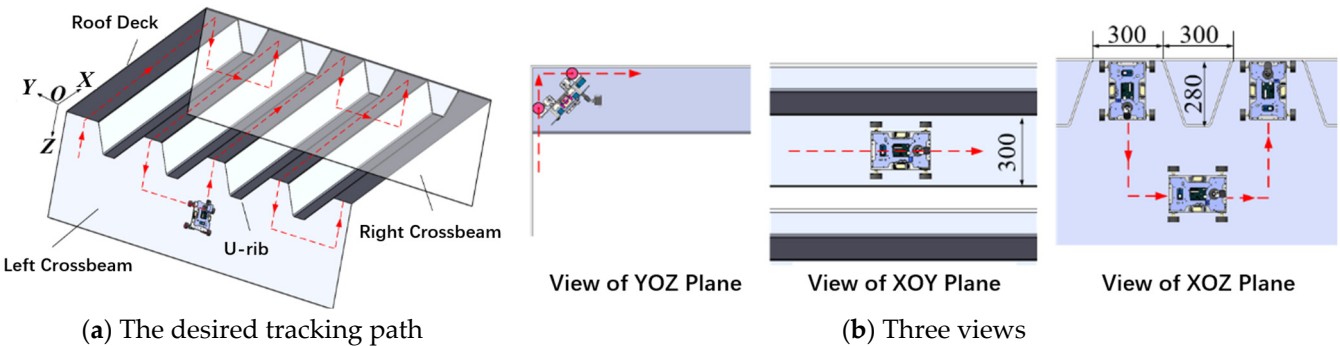

(**a**) The desired tracking path        (**b**) Three views

**Figure 4.** Schematic of the robot inspecting the SBG.

### 3.1. Crossing a 90° Concave Corner

The analysis is based on the example of the robot turning over from the left crossbeam to the roof deck. This process can be divided into the following stages: (1) the robot moves forward until its front wheels touch the roof; (2) the lifting mechanism lifts the robot to separate the front wheel from the left crossbeam; (3) the front wheel of the robot moves on the roof, while the rear wheel moves on the left crossbeam. It continues to go forward until the entire vehicle is attached to the roof.

The action of the lifting mechanism is shown in Figure 5. The lifting mechanism starts to rotate when the front laser rangefinder detects that the front wheels are touching the roof deck. The front wheel is lifted to a maximum distance when the lifting arm rotates perpendicular to the left crossbeam. The lifting mechanism is then reset.

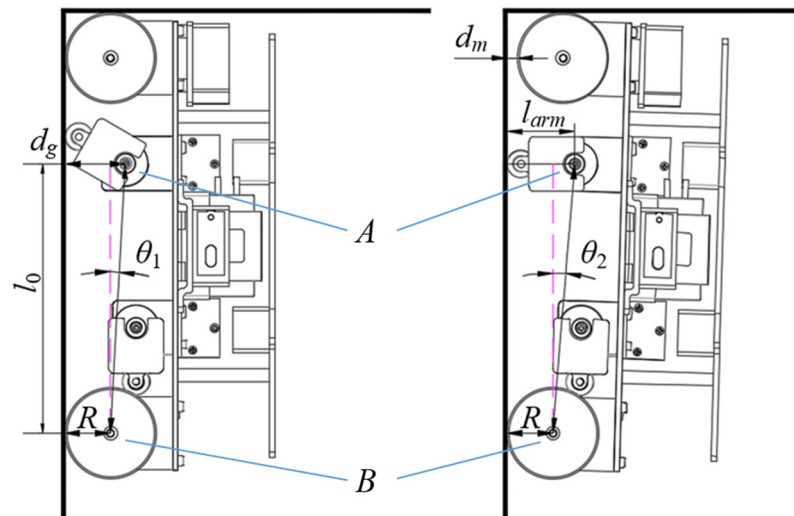

**Figure 5.** Schematic of the robot crossing a 90° concave corner.

Point *A* is the rotation center of the lifting mechanism, and point *B* is the center of the rear wheel. It can be seen from Figure 5 that when the lifting mechanism is reset, the pitch angle change of the robot can be calculated as follows:

$$\Delta\theta = \theta_2 - \theta_1$$
$$= \arcsin\frac{l_{arm}-R}{|AB|} - \arctan\frac{d_g-R}{l_0}, \tag{1}$$

where

$$|AB| = \sqrt{l_0{}^2 + (d_g - R)^2}, \tag{2}$$

and

$\theta_1$—the angle between line *AB* and the left crossbeam when the lifting mechanism starts to rotate,

$\theta_2$—the angle between line *AB* and the left crossbeam when the lifting mechanism is reset,

$l_0$—the vertical distance between *A* and *B*,

$l_{arm}$—the arm length of the lifting mechanism,

$d_g$—the vertical distance from *A* to the left crossbeam,

*R*—the wheel radius.

Figure 6 shows the process of the front and rear wheels moving on the roof and the left crossbeam, respectively. In this process, the ratio of front and rear wheel speeds changes with the change of the robot's pitch angle $\theta$.

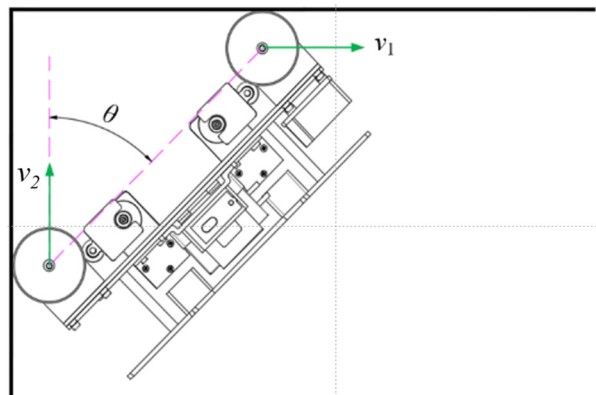

**Figure 6.** Relationship between the front and rear wheel speeds while crossing the corner.

The front and rear wheel speeds $v_1$ and $v_2$ satisfy the following formula:

$$v_2 = v_1 \tan \theta. \tag{3}$$

The actual speeds of the front and rear wheels have upper limits. Therefore, the following speed control formula is proposed:

$$\begin{cases} v_2 = v_1 \tan \theta \, ; \; v_1 = 20\text{mm/s} \, , \; \theta \in \left(0, \frac{\pi}{4}\right] \\[2mm] v_1 = v_2 / \tan \theta \, ; \; v_2 = 20\text{mm/s} \, , \; \theta \in \left[\frac{\pi}{4}, \frac{\pi}{2}\right) \\[2mm] v_2 = v_1 = 20\text{mm/s} \, , \; \theta = \left\{0, \frac{\pi}{2}\right\} \end{cases} . \tag{4}$$

### 3.2. Tracking the Lane's Centerline

While detecting the weld between the U-rib and the roof in the lane (Figure 7), the robot's desired path is the centerline of the lane. However, the commonly used IMU has a dynamic drift phenomenon, which will cause errors in the measurement [21]. At present, most researchers use magnetometer [22] or GPS [23] to correct this error; however, the magnetic field around the magnet wheel is so strong that the magnetometer won't operate normally. In addition, the highly enclosed space in the SBG limits the application of GPS.

To overcome the difficulties in measuring the tracking error, a trilateration localization algorithm is proposed. This algorithm uses the distance information measured by laser rangefinders $l_2$, $l_3$, and $l_4$ to calculate $d$ and $\delta$. $d$ is the distance deviation between the robot center $O_c$ and the lane's centerline, and $\delta$ is the yaw deviation between its speed direction and the centerline. In this paper, the heading direction of the robot is defined as the positive direction of the speed. This paper also stipulates that when $O_c$ is on the right side of the centerline, $d > 0$; when the robot heads to the left of the centerline, $\delta > 0$ (as shown in Table 1).

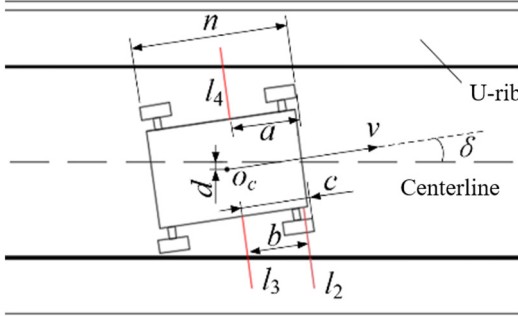

**Figure 7.** Schematic of the robot inspecting the lane.

**Table 1.** The sign of $\delta$ and $d$.

| | $d > 0$ | $d < 0$ |
|---|---|---|
| $\delta > 0$ | $O_c$ is on the right side of the centerline, and the robot heads to the left of the centerline. | $O_c$ is on the left side of the centerline, and the robot heads to the left of the centerline. |
| $\delta < 0$ | $O_c$ is on the right side of the centerline, and the robot heads to the right of the centerline. | $O_c$ is on the left side of the centerline, and the robot heads to the right of the centerline. |

From Figure 7,

$$\begin{cases} \delta = \tan^{-1}\left(\frac{l_2 - l_3}{b}\right) \\ d = \frac{(l_4 - l_3)\cos\delta + (n - a - c)\sin\delta}{2} \end{cases} , \tag{5}$$

where

$a$—the distance from laser rangefinder $l_4$ to the front wheel,
$b$—the distance between laser rangefinder $l_2$ and $l_3$,
$c$—the distance from laser rangefinder $l_3$ to the front wheel,
$n$—the length of the wall-climbing robot.

With the tracking error obtained, a "Preview-follow" algorithm [24,25] is adopted to track the desired path according to the low-speed working conditions (as shown in Figure 8). When deviating from the centerline path, the algorithm controls the robot to move along arc $S_0$ from $O_c$ to point $P$, and then follow the centerline path. $P$ is the intersection point of arc $S_0$ and the centerline, which is called the preview point.

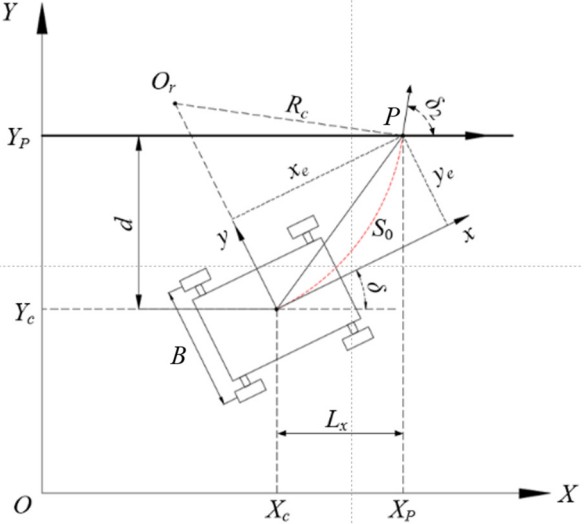

**Figure 8.** "Preview-follow" path tracking algorithm.

A coordinate system {$XOY$} is established on the roof. Its $X$-axis is parallel to the centerline, and its $Y$-axis is perpendicular to the centerline. We define the local coordinate system of the wall-climbing robot as coordinate system {$XOY$}, and define the heading direction of the robot as the positive direction of the $x$-axis. The $y$-axis is perpendicular to the heading direction. From Figure 8,

$$\begin{cases} x_e{}^2 + y_e{}^2 = L_x{}^2 + d^2 \\ (R_c - y_e)^2 + x_e{}^2 = R_c{}^2 \\ x_e = d \sin \delta + L_x \cdot \cos \delta \\ y_e = d \cos \delta - L_x \cdot \sin \delta \end{cases}, \tag{6}$$

where

$x_e$—the abscissa of point $P$ in coordinate system {$XOY$},
$y_e$—the ordinate of point $P$ in coordinate system {$XOY$},
$\delta$—the yaw deviation,
$L_x$—the horizontal distance between $O_c$ and $P$, which is called the preview distance,
$R_c$—the turning radius of $O_c$.

From Formula (6), we can obtain

$$R_c = \frac{L_x{}^2 + d^2}{2(d \cos \delta - L_x \sin \delta)}. \tag{7}$$

When the wall-climbing robot reaches the preview point $P$, from Figure 8, we can obtain

$$\delta_2 = \arccos(\frac{R_c \cos \delta - d}{R_c}), \tag{8}$$

where

$\delta_2$—the angle between speed direction and the centerline path.

Then, the robot rotates around $O_c$ in situ until its yaw deviation to the centerline is zero. During this process, the speeds of the left and right wheels are:

$$\begin{cases} v_r = -\text{sign}(\delta)\frac{\omega B}{2} \\ v_l = \text{sign}(\delta)\frac{\omega B}{2} \end{cases}, \tag{9}$$

where

$v_l$—the speed of the left wheel,
$v_r$—the speed of the right wheel,
$B$—the distance between the left and right wheels.

### 3.3. Steering to an Adjacent Lane

Without the Ackerman steering mechanism, obvious slippage occurs when the robot turns. Confronting the slippage problem, the desired path is preferably composed of straight lines, which makes it easier to measure the yaw deviation and then control the tracking errors. Therefore, a U-shaped path composed of three straight lines is designed for the robot to steer into an adjacent lane, as shown in Figure 9. The following analysis takes the robot steering into an adjacent lane on the left as an example.

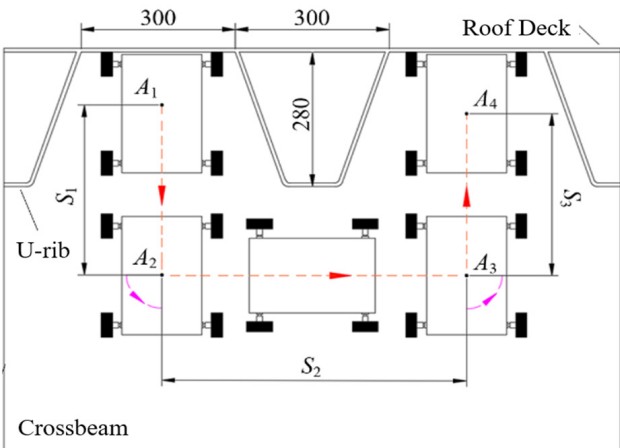

**Figure 9.** Desired tracking path of the robot steering to adjacent lane.

The wall-climbing robot travels along the path $S_1$—$S_2$—$S_3$, which can be divided into the following processes: (1) the robot travels straight down from $A_1$ to $A_2$ for a distance of $S_1$, at which time it drives away from the original lane; (2) the robot turns left 90° at $A_2$; (3) the robot travels straight ahead from $A_2$ to $A_3$ for a distance of $S_2$, at which time $O_c$ is located on the centerline of the next lane; (4) the robot turns left 90° at $A_3$; (5) the robot travels straight up from $A_3$ to $A_4$ for a distance of $S_3$, at which time it has entered the next lane.

Figure 10 shows the pose of the robot steering into the next lane on the crossbeam. This process uses the encoder/IMU system to perform dead reckoning. The distance traveled on three straight lines can be calculated through the encoder. However, slippage during steering will cause a significant error in the yaw angle calculated by the dead-reckoning. Therefore, this paper uses IMU to monitor the angle change of the robot when steering in situ on the crossbeam.

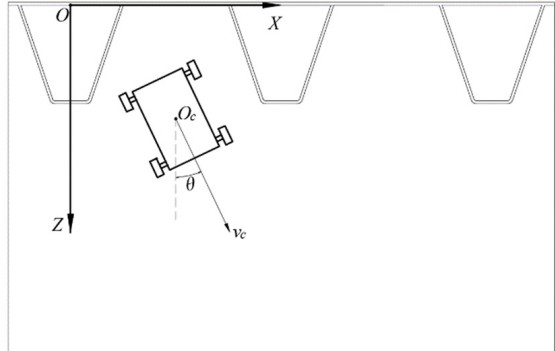

**Figure 10.** Pose of the robot steering to adjacent lane.

The dead reckoning formula is:

$$\begin{cases} X_{i+1} = X_i + \Delta X \\ Z_{i+1} = Z_i + \Delta Z \\ \theta_{i+1} = \theta_i + \Delta \theta \end{cases}, \tag{10}$$

where $[X_{i+1}, Z_{i+1}, \theta_{i+1}]^{\mathrm{T}}$ and $[X_i, Z_i, \theta_i]^{\mathrm{T}}$ represent the robot's poses at $(i + 1)$-th and $i$-th sampling moments, respectively; and $\Delta X$, $\Delta Y$, and $\Delta \theta$ represent the amount of change in pose between these two moments.

The relationship between pose change and wheel speed can be expressed as:

$$\begin{cases} \Delta X = \frac{v_r + v_l}{2} \sin \theta_i \cdot \Delta T \\[2mm] \Delta Z = \frac{v_r + v_l}{2} \cos \theta_i \cdot \Delta T \quad . \\[2mm] \Delta \theta = \frac{v_l - v_r}{B} \cdot \Delta T \end{cases} \tag{11}$$

The position of the robot in three straight lines can be calculated through the encoder, but the slippage in the steering process will lead to a relatively large error in the yaw angle obtained by dead reckoning. Therefore, this article uses an IMU to obtain the yaw angle of the robot during steering.

## 4. Experiment

This section discusses the experiments undertaken on crossing 90° concave corner, tracking the lane's centerline, and steering into the adjacent lane to analyze the control performance.

### 4.1. Experiment on Crossing a 90° Concave Corner

Stable crossing of a 90° concave corner is essential for the wall-climbing robot to complete path tracking. To study the stability of the robot crossing the corner, some experiments were carried out to test the load capacity and the smoothness of the action. Figure 11a,b shows the process of the robot crossing the corner with and without a 4.3 kg load, respectively. Figure 12 shows the time history of the robot's pitch angle $\theta$ while crossing a 90° concave corner. The curve slope does not have an abrupt change, which shows that the crossing process is stable. Given that the robot and camera weigh only 4.08 kg, there will still be sufficient margin for the load capacity with a safety factor of 1.5. The experiment shows that the wall-climbing robot can stably cross the corner with the required equipment.

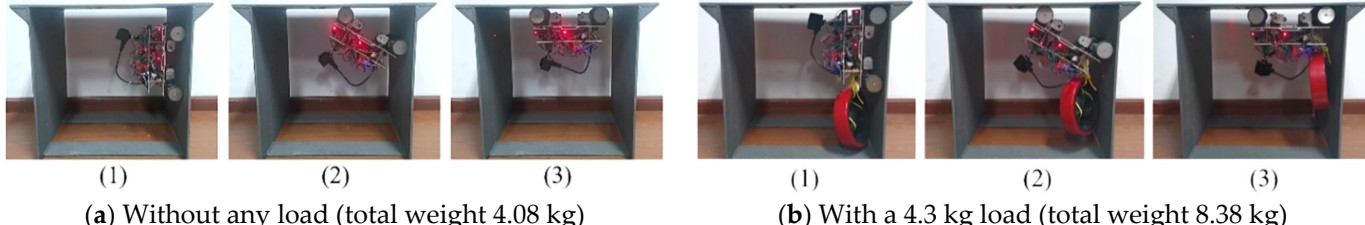

(1)    (2)    (3)        (1)    (2)    (3)

(**a**) Without any load (total weight 4.08 kg)    (**b**) With a 4.3 kg load (total weight 8.38 kg)

**Figure 11.** Experiment of the robot crossing a 90° concave corner.

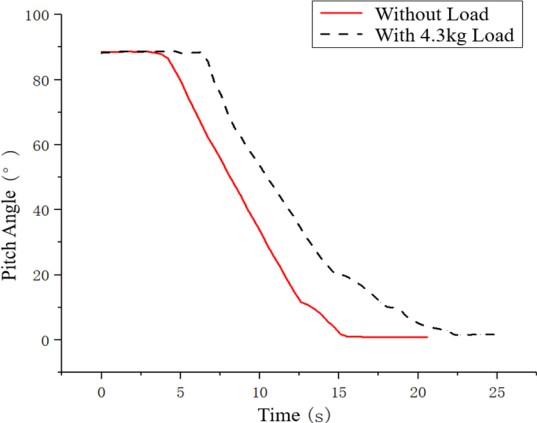

**Figure 12.** Time history of the robot's pitch angle while crossing a 90° concave corner.

### 4.2. Experiment on Tracking the Lane's Centerline

According to the path tracking analysis in Section 3, the preview distance $L_x$ is the parameter to be determined of the "Preview-follow" algorithm, which determines the horizontal distance left for the robot to correct its pose. To verify the effectiveness of the "Preview-follow" algorithm and determine the appropriate preview distance, this section carries out the path tracking experiments with different preview distances $L_x$, as shown in Figure 13.

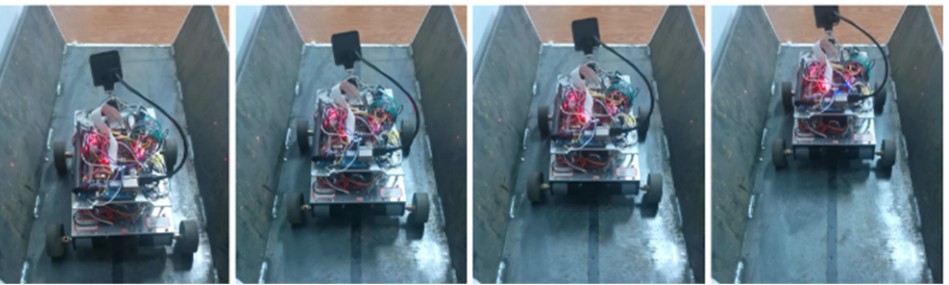

**Figure 13.** Experiment of the robot tracking the lane's centerline.

Figure 14 shows the time history of $d$ and $\delta$ during the experiment. It can be seen from the figure that the robot can track the centerline with or without preview distance ($L_x > 0$ or $L_x = 0$). When $L_x = 20$ mm, the wall-climbing robot responds faster in tracking the centerline path, and the tracking error is also smaller. The final distance deviation and yaw deviation of the tracking are 0.33 mm and 0.06°, respectively. Therefore, $L_x = 20$ mm is chosen as the control parameter of centerline path tracking.

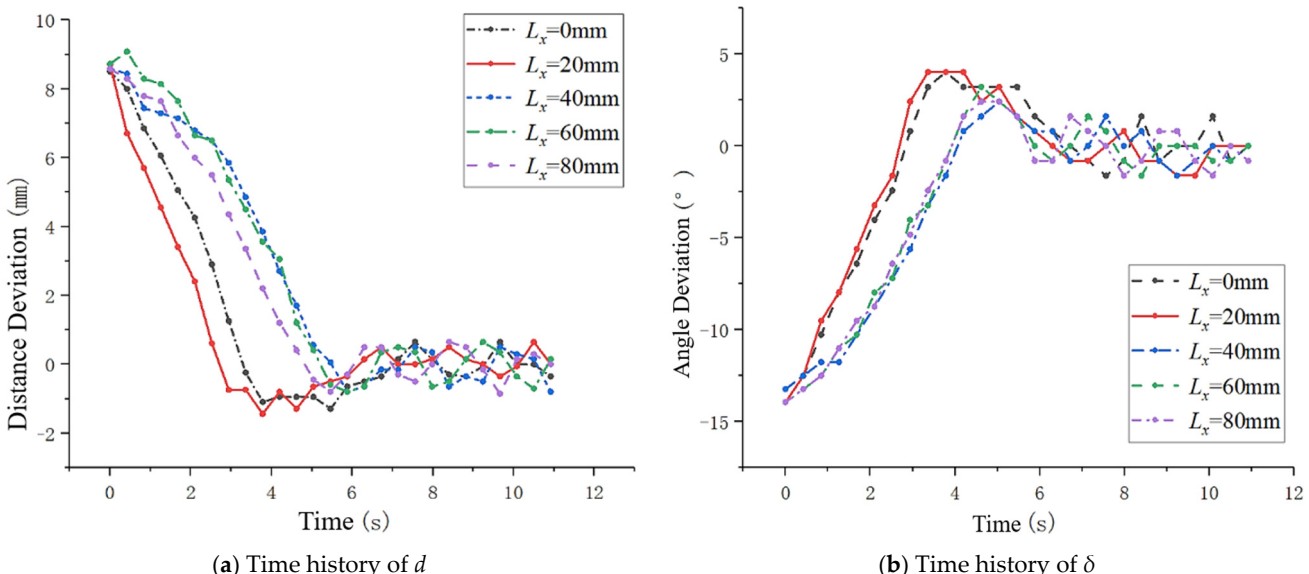

(a) Time history of $d$

(b) Time history of $\delta$

**Figure 14.** Error of the robot tracking the lane's centerline.

### 4.3. Experiment on Steering into an Adjacent Lane

Figure 15 is the experiment in which the wall-climbing robot tracks a U-shaped path. The experiment shows that the wall-climbing robot can track the proposed U-shaped path and steer into an adjacent lane. In this process, the robot moves smoothly and accurately. Furthermore, when it reaches the roof deck, the yaw deviation is less than 0.1°, which is proven by experiments to be small enough to not affect the robot's transition to the vertical wall.

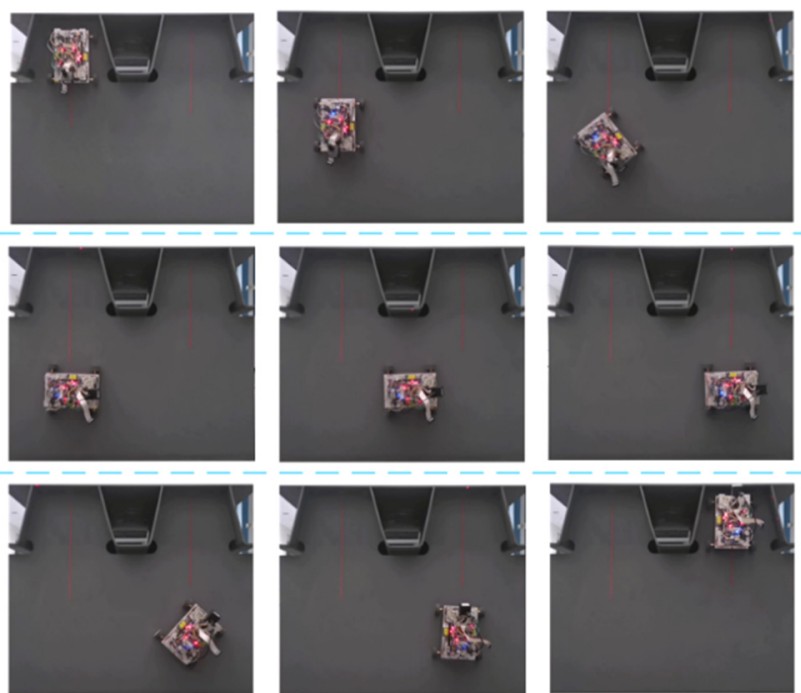

**Figure 15.** Experiment of the robot steering to next lane.

## 5. Conclusions

This paper proposes an autonomous inspection method using a wall-climbing robot for SBG. The method proposes a desired tracking path with inspection focuses covered for the wall-climbing robot to obtain the images inside SBG. Then the method adopts several path tracking algorithms for the three typical actions during inspection to realize stable and efficient movements.

The desired path is composed of the lane's centerline and a U-shaped steering path. In tracing the lane's centerline, a trilateration localization method is designed to obtain the robot's pose. A path tracking algorithm based on the "Preview-follow" theory is adopted to optimize the tracking effect. After experiments, an optimal preview distance was selected to make the wall-climbing robot track the centerline faster and with a smaller tracking error.

In steering, straight paths combined with in-situ steering are adopted to restrain slippage. Then, the dead reckoning algorithm is adopted to achieve tracking of the desired path. Experiments show that the algorithm can effectively control the tracking error, making the yaw deviation close to 0 at the beginning of the transition. In addition, to achieve a smooth transition between roof and crossbeam, an algorithm of cooperatively controlling the lifting mechanism and the wheel speeds is proposed.

In conclusion, the experiment results show that the wall-climbing robot can cross 90° concave corners steadily and track the desired path with high accuracy and fast response (including tracking the centerline and steering). The method proposed in this paper meets the autonomous SBG inspection requirements for the wall-climbing robot.

In the future, we will introduce dynamic analysis rather than ignoring the velocity change of the robot in this paper. Neglect of dynamics can lead to instability and inaccuracy of robot control in some cases of high speed. Further study could consider how to speed up inspection while ensuring the safety and accuracy of robot motion.

**Author Contributions:** Conceptualization, W.S., D.J. and S.Z.; methodology, Z.W.; software, Z.W.; validation, W.S., Z.W., T.W. and D.J.; formal analysis, T.W.; investigation, W.S.; resources, W.S.; data curation, W.S.; writing—original draft preparation, Z.W.; writing—review and editing, T.W.; visualization, Z.W.; supervision, W.S. and S.Z.; project administration, W.S. and S.Z.; funding acquisition, W.S. and S.Z. All authors have read and agreed to the published version of the manuscript.

**Funding:** This work is supported by Open Project of Technology and Equipment of Rail Transit Operation and Maintenance Key Laboratory of Sichuan Province (Grant No. 2019YW002), Quality Technology Infrastructure Project of Zhejiang Administration for Market Regulation (Grant No. 20200133) and Eaglet Planning Cultivation Project of Zhejiang Administration for Market Regulation (Grant No. CY2022231).

**Institutional Review Board Statement:** Not applicable.

**Informed Consent Statement:** Not applicable.

**Data Availability Statement:** Not applicable.

**Conflicts of Interest:** The authors declare no conflict of interest.

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
