# Peer review of "A Path Tracking Method of a Wall-Climbing Robot towards Autonomous Inspection of Steel Box Girder"

_machines, doi:10.3390/machines10040256_

Round 1

Reviewer 1 Report

Dear Authors

This is an interesting article about a wall climbing inspection robot. It is related to the important issue of inspection in hard to reach places.

The main part of the article is described in detail, but there are some minor issues that require clarification in order to improve the readability of the paper.

The introduction does not provide enough information and the literature review should be improved.

The prototype uses magnetic wheels, which are described very briefly. They allow you to climb and move over the ceiling, but there are some limitations of this solution that should be mentioned in the article, as it may be of interest to readers.

The assumptions and simplifications of the model should be clearly described.

The robot dynamics model of has been completely omitted (It may be the topic of further research).

The slippage problem is important, as it affects the positioning performance of the wheeled mobile robots and can cause a navigational error, therefore it should be described in more detail.

The conclusions are very brief. It should be improved and should include limitations of the presented method.

A spell check of the English language is required.

Reviewer 2 Report

This paper proposes a method for navigating an autonomous wall-climbing robot towards autonomous inspection of steel box girder. The application that the robot is indented to be used is very interesting. However, there are several issues that should be addressed:

  • The title is misleading. The paper is focused on the tracking method of the robot entirely.
  • The introduction as well as the contribution part of the paper should be in-line with the remaining text.
  • The paper has a lot of scientific errors and more specific:
    • The reference system in Fig4a is left-handed.
    • Eqs(2) and (3) are wrong. The velocities v_1 and v_2 of points A and B respectively, should relate the angular velocity of the robot body. Please advise the textbook “Vector mechanics for engineers” by Beer and Johnston.
    • Since the sign of \lamda and d is important use vectors to specify them in fig.7.
    • The symbols in figure 7 does not follow the description of them presented after eq (4).
  • Eq(1) and Eq (4) are big without describing why (e.g.\theta2=arcsin((l_arm-R)/l_0)

  • I don’t understand why the desired path composed by straight paths restrains the slippage.
  • Is the safety factor correct? (My calculation is equal to 1,05).
  • Why Lx is so important in the definition of distance deviation. What is the meaning of Lx=0. Is the orientation of the robot important?
  • In the introduction avoid referring to Universities. Specify the research groups. In addition, avoid the word “literature”.

Reviewer 3 Report

Well done.

The paper is well written, clear, concise and easy to read. The contributions highlighted in the conclusions are supported by experimental results, collected using a robot equipped with sensors.
The main comments I have regard the decision to neglect the dynamics of the system in the model and the lack of experimental results representatives of the robot's path throughout the entire application (inspection of SGB of a bridge). Expanding on these two areas would strengthen the content of the paper, however this would also increase the length of the article, thus they may be considered for future work.

Round 2

Round 3

Reviewer 2 Report

None